# Clinical Significance of Rotational Thromboelastometry (ROTEM) for Detection of Early Coagulopathy in Trauma Patients: A Retrospective Study

**DOI:** 10.3390/diagnostics15091148

**Published:** 2025-04-30

**Authors:** Mohammad Asim, Ayman El-Menyar, Ruben Peralta, Suresh Arumugam, Bianca Wahlen, Khalid Ahmed, Naushad Ahmad Khan, Amani N. Alansari, Monira Mollazehi, Muhamed Ibnas, Ammar Al-Hassani, Ashok Parchani, Talat Chughtai, Sagar Galwankar, Hassan Al-Thani, Sandro Rizoli

**Affiliations:** 1Clinical Research, Trauma and Vascular Surgery, Hamad Medical Corporation, Doha P.O. Box 3050, Qatar; masim1@hamad.qa (M.A.); naushadkhan82@gmail.com (N.A.K.); muhamedibnas@gmail.com (M.I.); 2Clinical Medicine, Weill Cornell Medical College, Doha P.O. Box 3050, Qatar; 3Department of Surgery, Trauma Surgery, Hamad Medical Corporation, Doha P.O. Box 3050, Qatar; rperaltamd@gmail.com (R.P.); aarumugam@hamad.qa (S.A.); kahmed11@hamad.qa (K.A.); mollazehi@hamad.qa (M.M.); ammar_alhassani@yahoo.com (A.A.-H.); aparchani@yahoo.com (A.P.); tchughtai@hamad.qa (T.C.); srizoli@hamad.qa (S.R.); 4Department of Surgery, Universidad Nacional Pedro Henriquez Urena, Santo Domingo P.O. Box 1423, Dominican Republic; 5Department of Anesthesiology, Hamad Medical Corporation, Doha P.O. Box 3050, Qatar; bwahlen@googlemail.com; 6Department of Pediatric Surgery, Hamad Medical Corporation, Doha P.O. Box 3050, Qatar; aalansari9@hamad.qa; 7Emergency Medicine Residency Program, College of Medicine, Sarasota Memorial Hospital, Florida State University, Sarasota, FL 34243, USA; gcsagar@yahoo.com; 8Department of Surgery, Trauma and Vascular Surgery, Hamad Medical Corporation, Doha P.O. Box 3050, Qatar; althanih@hotmail.com

**Keywords:** polytrauma, coagulation abnormalities, ROTEM, severity of injury, outcomes

## Abstract

**Background**: We aimed to evaluate the clinical significance of abnormal rotational thromboelastometry (ROTEM) findings in trauma patients and investigate the relationships between FIBTEM-maximum clot firmness (MCF), fibrinogen concentration and patient outcomes. **Methods**: A retrospective cohort analysis was conducted on adult trauma patients who underwent on-admission ROTEM testing between January 2020 and January 2021. Univariate analyses compared data based on injury severity, ROTEM findings (normal vs. abnormal), and initial fibrinogen concentration (normal vs. hypofibrinogenemia). ROC curve analysis was performed to determine the diagnostic performance of FIBTEM A10/MCF for its association with hypofibrinogenemia. **Results**: A total of 1488 patients were included in this study; the mean age was 36.4 ± 14.2 years and 92% were male. In total, 376 (25.3%) patients had ROTEM abnormalities. Severe injuries (ISS ≥ 16) were associated with a higher shock index, positive troponin T levels, standard coagulation abnormalities, hypofibrinogenemia, and abnormal ROTEM parameters (*p* < 0.05). These patients also had higher rates of massive transfusions and in-hospital mortality (*p* = 0.001). Coagulation alterations were significantly associated with higher injury severity score (ISS), shock index, head abbreviated injury score (AIS), hypofibrinogenemia, transfusion need, and mortality (*p* < 0.05). Hypofibrinogenemic patients were younger, sustained severe injuries, had higher shock indices and coagulation marker levels, required more intensive treatments, had longer hospital stays, and had higher mortality (*p* < 0.05). A significant positive correlation was found between plasma fibrinogen concentration and FIBTEM-MCF (r = 0.294; *p* = 0.001). **Conclusions**: Approximately one-fourth of the patients had early traumatic coagulopathy, as assessed by ROTEM. The FIBTEM A10/MCF may serves as a surrogate marker for plasma fibrinogen concentration. While prior studies have established the link between ROTEM and injury severity, our findings reinforce its relevance across varying trauma severity levels. However, prospective studies are warranted to validate its role within diverse trauma systems and evolving resuscitation protocols.

## 1. Introduction

Trauma is the primary cause of death and disability among young people and remains a substantial public health concern worldwide [1,2]. In severely injured patients, both acute bleeding and trauma-induced coagulopathy (TIC) are major factors responsible for 25% of all trauma-related deaths [3]. For effective trauma care, the early identification and management of hemorrhage and TIC—a condition that includes and exacerbates hemorrhagic tendencies—is of paramount importance, both in the prehospital setting and during initial resuscitation in the hospital [4]. The most widely accepted hemostatic resuscitation principle for patients with significant hemorrhage mandates predetermined blood transfusion ratios (1:1:1) followed by goal-directed strategies that rely on laboratory findings with long turnaround times [5]. However, delays in accessing coagulation lab tests imposes notable limitations, that may lead to adverse outcomes [6]. Viscoelastic testing methods such as rotational thromboelastometry (ROTEM), have been shown to be advantageous compared to conventional coagulation tests and are more capable of early detection of TIC [7,8]. Furthermore, severely injured patients, as indicated by an injury severity score (ISS) ≥ 16 [9], are at increased risk of bleeding and TIC, requiring hemostatic resuscitation [10,11].

In recent years, ROTEM has been utilized as a point-of-care (bedside) test in trauma patients, providing quick results without the need for pre-analytic sample preparation [12]. Whole-blood samples can be evaluated by employing four distinct measurement modalities, i.e., EXTEM for assessing the extrinsic pathway, INTEM for the intrinsic pathway, FIBTEM for determining fibrinogen function, and APTEM for fibrinolysis detection [13]. The prompt diagnosis of TIC by ROTEM is beneficial in managing bleeding trauma patients [14]. Among the most useful information ROTEM can provide is fibrinogen activity. Fibrinogen, the precursor of fibrin, is an essential component of healthy blood clots. Fibrinogen/fibrin binds to αIIbβ3 integrins of already activated platelets and leads to the aggregation of platelets, thereby stabilizing the clot [15]. Current evidence indicates that a low admission fibrinogen concentration is independently associated with injury severity and mortality. Furthermore, FIBTEM-ROTEM is a rapid means of assessing hypofibrinogenemia in trauma patients [16]. Therefore, understanding the relationship between admission fibrinogen activity measured by ROTEM and injury severity could elucidate how early coagulation abnormalities manifest in relation to the extent of trauma. This study aimed to evaluate the clinical significance of abnormal ROTEM findings in adult trauma patients and to demonstrate the correlation between FIBTEM-maximum clot firmness (MCF) and fibrinogen concentration and patient outcomes.

## 2. Materials and Methods

Methods: A retrospective analysis was conducted for all adult trauma patients who underwent point-of-care ROTEM upon admission to the Hamad Trauma Center (HTC) in Qatar. Data were retrieved from the Qatar National Trauma Registry (QNTR) database and electronic medical records (CERNER) over a 12-month period starting in January 2020. Patients in the pediatric age group (<14 years) or those who did not undergo ROTEM viscoelastic testing upon hospital arrival were excluded.

The following data were collected from all patients: demographic information (age and sex), injury severity score (ISS), abbreviated injury score (AIS), revised trauma score (RTS) [17], routine laboratory findings on admission (international normalized ratio, prothrombin time, activated partial thromboplastin time, platelet count, blood pH, lactate, hematocrit, WBC count, and base excess and troponin T), blood transfusion (blood units transfused overall and within the first 24 h), massive blood transfusion, procedures (exploratory laparotomy, thoracotomy, craniotomy/craniectomy, open reduction internal fixation, chest tube insertion, and intubation), drug administration (prothrombin complex, fibrinogen, tranexamic acid, and vasopressors), in-hospital complications (ventilator-associated pneumonia, sepsis, acute respiratory distress syndrome [ARDS], and multiorgan failure), mechanical ventilation, intensive care unit (ICU) and hospital length of stay, and mortality. ROTEM analysis was performed upon arrival at the hospital in the Trauma Resuscitation Unit. As soon as intravenous access was established, routine blood samples were collected for initial bloodwork, including ROTEM.

This study analyzed the extrinsically activated assays of viscoelastic measurements, i.e., ROTEM (EXTEM and FIBTEM), which included the EXTEM clotting time (CT), which measures the time to initial clot formation (speed of coagulation activation); EXTEM clot firmness (amplitude) at 10 min (A10), which indicates the clot strength at 10 min, providing an early assessment of clot quality; EXTEM-maximum clot firmness (MCF), which represents the overall strength and stability of the clot, reflecting contributions from fibrin, platelets, and clotting factors; EXTEM maximum lysis (ML), which measures the degree of clot breakdown; FIBTEM-CT, which reflects the time to fibrin-based clot formation, specifically fibrinogen activity; and FIBTEM-A10/MCF, which determines the role of fibrinogen to clot firmness, guiding the need for fibrinogen replacement therapy [18]. The normal ranges of the ROTEM parameters were established by the manufacturer (ROTEM^®^ Sigma; Werfen, Barcelona, Spain) and the measurements were performed as described earlier by Schenk et al. [19]. We measured plasma fibrinogen concentration using the Clauss method (Turbidimetric), assessed by using different analyzer platforms for direct/indirect immunofluorescence assays.

The presence of TIC was based on the identification of any abnormality in one or more of the ROTEM parameters. This approach is consistent with the methodology used by Bonet et al. [20], who defined hypocoagulability using ROTEM parameters, considering acute trauma coagulopathy as present in patients with any ROTEM values outside the normal range. Severe trauma was defined as an injury severity score (ISS) greater than 15 [17]. Hypofibrinogenemia was defined as a concentration of initial fibrinogen ≤ 2.0 g/L [21]. A massive blood transfusion was defined as the transfusion of more than 10 units of packed red blood cells within 24 h [22]. As there was no direct contact with the patients and data were collected anonymously, this study was exempted and approved by the Institutional Review Board of Hamad Medical Corporation, Qatar (MRC-01-21-078, dated 14 February 2021). All methods were performed in accordance with the relevant guidelines and regulations. This study follows the STROBE checklist for observational studies.

Statistical analysis: Descriptive variables are presented as proportions, means ± standard, medians and 95% confidence intervals, as appropriate. The Kolmogorov–Smirnov test was used to assess normality. Comparative analyses were performed according to the severity of injury, categorized by ISS [mild (ISS ≤ 8), moderate (ISS 9–15), and severe (ISS ≥ 16)], Glasgow Come Score [moderate (GCS 9–12) and severe (GCS ≤ 8)], traumatic brain injury (TBI), ROTEM findings (normal vs. abnormal) and initial fibrinogen concentration (non-hypofibrinogenemia vs. hypofibrinogenemia). Differences in categorical variables were examined using the chi-square test or Fisher’s exact test. Continuous variables were compared using Student’s *t* test for two groups or ANOVA for >two groups for parametric data. The Mann–Whitney U test and Kruskal–Wallis test were used for nonparametric data, where applicable. To assess the relationship between viscoelastic ROTEM parameters and plasma fibrinogen levels, bivariate correlation analysis was performed using Pearson’s correlation coefficient. This was applied specifically to evaluate the association between FIBTEM-MCF and plasma fibrinogen concentration (measured by the Clauss method). Receiver operating characteristic (ROC) curve analyses was performed to determine the diagnostic performance of FIBTEM-MCF/A-10 and/or EXTEM-CT/A10 for association with hypofibrinogenemia, need for blood transfusion, massive transfusion, and in-hospital mortality. An area under the curve (AUC) of 1.0 was considered to indicate perfect discrimination and 0.5 was considered to indicate equal chance. A two-tailed *p* value less than 0.05 was considered to indicate statistical significance. The data were analyzed using the Statistical Package for the Social Sciences version 22.0 (SPSS, Inc., Chicago, IL, USA).

## 3. Results

During the 12-month period, a total of 1488 adult patients were admitted to the HTC, and ROTEM analysis was performed upon hospital admission. A descriptive analysis of the study cohort is presented in Table 1. The mean age of the patients was 36.4 ± 14.2 years, the majority were male (91.8%), and most had sustained blunt trauma (96.2%). The median GCS score was 15 (3–15), the mean shock index was 0.71 ± 0.3, the ISS was 11.5 ± 8.6, with 24% of the patients being severely injured (ISS ≥ 16). Approximately one-fourth (25.3%) of patients had abnormal ROTEM results, meeting our definition of having TIC. Abnormal FIBTEM and EXTEM findings were observed in 10.9% and 20.8% cases, respectively. Figure 1 shows an example of ROTEM (FIBTEM and EXTEM) parameters in trauma patients. Overall, 14% of patients received blood transfusions, with a median of 4 units (range 1–42), and 2% required massive transfusions within the first 24 h. Thirteen percent of patients required mechanical ventilation. The median ICU length of stay was 3 days (0.04–74) and the median hospital stay was 4 (1–366) days. The overall mortality rate was 2.2% (33 patients).

Table 2 presents a comparative analysis of clinical characteristics and ROTEM findings based on injury severity scores. Patients with severe injuries (ISS ≥ 16) were predominantly male (*p* = 0.009) and had a significantly greater mean shock index (*p* = 0.001), median troponin T level (*p* = 0.001), prothrombin time (*p* = 0.001), international normalized ratio (INR) (*p* = 0.001), lactate level (*p* = 0.001), base excess (*p* = 0.001), and lower fibrinogen concentration (*p* = 0.01) than did the other groups. Moreover, the incidence of hypofibrinogenemia (*p* = 0.001) was significantly greater in the severe injury group. Regarding ROTEM parameters, specifically the median FIBTEM A10 (*p* = 0.001), FIBTEM-MCF (*p* = 0.001), EXTEM A10 (*p* = 0.001), and EXTEM-MCF (*p* = 0.001), were significantly lower in patients with severe injuries. Furthermore, the need for blood transfusion (*p* = 0.001), massive transfusion (*p* = 0.001) and the rate of in-hospital mortality (*p* = 0.001) were greater among severely injured patients.

Table 3 shows a sub-analysis to illustrate the difference between moderate and severe TBI based on the GCS score. The two groups were comparable in terms of age, sex, and routine laboratory parameters. Patients with severe TBI were more likely to have higher EXTEM CT (*p* = 0.03) and EXTEM CFT (*p* = 0.01) values and a greater rate of in-hospital mortality (*p* = 0.02) than were those with moderate TBI. However, the median EXTEM A10 was significantly higher in the moderate TBI group (*p* = 0.02).

Table 4 lists the clinical characteristics and outcomes of patients with and without thromboelastometry abnormalities (TIC or no TIC). Patients with TIC were older (*p* = 0.001) and received greater volumes of prehospital fluids (*p* = 0.01). Upon admission, this group also presented with higher PT (*p* = 0.001), aPTT (*p* = 0.001), INR (*p* = 0.001), and lactate levels (*p* = 0.01). Moreover, the group with coagulation alterations had a significantly greater mean ISS (12.6 ± 9.8 vs. 11.0 ± 8.2; *p* = 0.007), shock index (0.75 ± 0.32 vs. 0.69 ± 0.28; *p* = 0.003) and head AIS (3.5 ± 1.0 vs. 3.2 ± 0.9; *p* = 0.04). The rates of hypo-fibrinogenemia (*p* = 0.001), need for blood transfusion within 24 h and total units transfused (*p* = 0.001), administration of fibrinogen (*p* = 0.001) and vasopressors (*p* = 0.001), and in-hospital mortality were significantly greater in patients with abnormal ROTEM profiles.

Table 5 presents a comparative analysis of the initial fibrinogen concentrations of trauma patients. Patients with hypofibrinogenemia were younger (*p* = 0.01), had sustained severe injuries (*p* = 0.001), and had a greater shock index (*p* = 0.001), PT (*p* = 0.001), and INR (*p* = 0.001) than those with normal fibrinogen concentrations. The incidence of abnormal FIBTEM (*p* = 0.001) and overall ROTEM (*p* = 0.001) results were significantly greater in the hypofibrinogenemia group. Furthermore, these patients were more likely to require intubation (*p* = 0.001) and massive blood transfusions (*p* = 0.001), to stay longer in the hospital (*p* = 0.002), and have higher rate of mortality (*p* = 0.007). Bivariate correlation analysis revealed a significant positive correlation between plasma fibrinogen concentrations and FIBTEM-MCF measurements (r = 0.294; *p* = 0.001).

Figure 2 shows the ROC curve for the association of hypofibrinogenemia with FIBTEM-MCF, using a cutoff value of 9.5 (AUC = 0.776; 95% CI = 0.735–0.818, *p* = 0.001; sensitivity = 77.2%; specificity = 66.7%), and FIBTEM-A10, with a cutoff value of 8.5 (AUC = 0.772; 95% CI = 0.730–0.815, *p* = 0.001; sensitivity = 73.1%; specificity = 72.8%).

FIBTEM-A10 and FIBTEM-MCF (cutoff ~9.5 mm) demonstrated moderate predictive value for low fibrinogen (≤2 g/L; AUC 0.77–0.78), blood transfusion, and massive blood transfusion (AUC 0.56–0.69). EXTEM parameters performed poorly (AUC ≤ 0.60), except EXTEM-CT in predicting mortality (AUC 0.76). Clauss fibrinogen (≤2 g/L) outperformed ROTEM in predicting massive blood transfusion (AUC 0.86). Overall, FIBTEM-A10 aids fibrinogen deficit detection but lacks high prognostic accuracy for outcomes (Table 6).

## 4. Discussion

This study explores the viscoelastic testing (ROTEM) findings in a cohort of patients with traumatic injuries. Specifically, we analyzed the FIBTEM and EXTEM parameters, as they are particularly relevant to clotting factors deficiencies and fibrinogen activity in trauma settings. FIBTEM provides insight into fibrinogen’s role in clot formation, directly addressing fibrinogen deficiency, whereas EXTEM assesses the extrinsic pathway of coagulation, offering clinically relevant information about overall clotting function. In contrast, INTEM, although useful for monitoring heparin effects in certain contexts, is less critical in trauma-related coagulopathy, as most of our patients are young and not typically on anticoagulation therapy. There are several key findings of our study. One-fourth of the patients had coagulopathy identified by ROTEM upon admission. Our study showed that the most common abnormal ROTEM results were FIBTEM-MCF and EXTEM-A10, suggesting deficits in fibrinogen activity and clotting factors. These observations were supported by the additional measurement of serum fibrinogen concentrations using the Clauss method, where 34% of the TIC patients had fibrinogen concentrations less than 2 g/L (defined as hypofibrinogenemia in this study). It is reasonable to assume that some abnormal ROTEM findings may not be clinically relevant—as slight variations from the normal ranges may not have clinical relevance. However, in the strict scientific analysis, any result outside of the normal range was defined as TIC. Notably, the most abnormal ROTEM finding in patients with severe injuries was FIBTEM results, indicating impaired fibrinogen activity. This aligns well with known evidence that fibrinogen is the first coagulation factor to decrease following trauma [23]. It is also plausible to consider that slight drops in fibrinogen concentrations, particularly in the absence of ongoing hemorrhage, while meeting the definition of TIC, do not require intervention and have no clinical significance. Moreover, abnormal laboratory test results could simply indicate a physiological or even pathological response to injury (including inflammatory response) and not necessarily a medical condition such as TIC, which requires treatment. Thus, one-quarter of all injured patients, including those with minor or moderate injuries, had an abnormal coagulation test upon arrival at the hospital, which may or may not be of clinical significance. While 25% of all patients had an abnormal ROTEM, only 5% had an abnormal INR. These findings also corroborate many previous observations/studies showing that the INR, despite its broad use and availability, was not designed to diagnose coagulopathy in trauma patients and may have limited use in this setting. Considering the clinical fact that 14% of all patients receive blood transfusions, it is plausible to infer that 25% of TICs diagnosed by an abnormal ROTEM are closer to the “clinical decision to transfuse” than the only 5% diagnosed by the INR. Notably, INR is commonly used as a marker to assess coagulopathy, particularly in the context of monitoring warfarin therapy and liver dysfunction. However, its role as a surrogate marker for coagulopathy, especially in trauma, is limited and often debated.

It is important to consider that both the ROTEM and INR are laboratory tests that may (or may not) assist clinicians in making decisions. While these tests are not perfect, our study suggested that the ROTEM may be more clinically relevant than the INR, even though we recognize that this conclusion is not supported by the findings of the present study. The coagulopathy group exhibited higher injury severity, elevated SI, more profound abnormalities in the standard coagulation test results, and a greater mortality rate. Furthermore, one-quarter of the patients sustained severe injuries and had a higher percentage of patients with hypofibrinogenemia and a significantly lower median FIBTEM-A10 and FIBTEM-MCF.

Similarly to our findings, Ojuka et al. [24] reported that among 140 major trauma patients, 85.7% were male, with a mean age of 29.4 years and a coagulopathy rate of 52.1%. In our study, 25% of trauma patients were found to have early coagulation alterations based on ROTEM, aligning with the 25–35% prevalence of acute TIC reported in other studies [20,25,26]. TIC is a complex, multifaceted condition characterized by abnormal blood clotting, influenced by prehospital interventions, in-hospital resuscitation, and physiological responses to injury and hemorrhagic shock [27]. Over the past decade, trauma care has increasingly emphasized early diagnosis through viscoelastic testing, along with advances in tailored resuscitation techniques and pharmacological treatments [28]. Platelet mapping by thromboelastography indicated that severely injured trauma patients had reduced platelet stimulation compared to healthy volunteers, with the degree of impairment correlating with the severity of injury [29]. The associations between injury severity and acidosis and coagulopathy have been widely studied [30,31].

Consistent with our findings, Savioli et al. [31] observed a significant association between TIC and the ISS, the presence of head and abdominal trauma, and the shock index, suggesting a link between these factors. Notably, patients with coagulation abnormalities due to trauma often experience severe bleeding and impaired hemostasis, necessitating aggressive management strategies such as early blood transfusions, administration of clotting factors such as fibrinogen, and the use of hemostatic agents such as tranexamic acid [31,32]. The higher mortality rate observed in the coagulopathy group reflects the challenge of achieving adequate hemostasis and managing associated complications. The presence of coagulopathy is linked to a fourfold increase in mortality in patients with similar injury severities [33,34]. A previous randomized controlled trial compared mortality in patients who received blood transfusions based on ROTEM results versus those managed with conventional coagulation tests [35]. The study revealed a significantly lower rate of mortality in the ROTEM-guided group (19.6% vs. 36.4%). The correlation between injury severity and TIC has been well documented [31]. In our study, 24% of the patients sustained severe injuries. This finding suggested that the greater the trauma severity is, the greater the likelihood of developing coagulopathy [36]. In severely injured patients, the hemostatic response becomes overwhelmed, resulting in reduced fibrinogen activity and impaired clot formation [31], as reflected in ROTEM parameters. Hypofibrinogenemia and decreased FIBTEM values indicate compromised fibrinogen function and clot strength, respectively, which are critical for effective hemostasis [37]. This relationship underscores the importance of early, targeted hemostatic interventions in severely injured patients to improve outcomes.

Our results demonstrated a significant positive correlation between plasma fibrinogen concentrations measured by the Clauss method and FIBTEM-MCF values. This finding aligns with previous studies showing that ROTEM parameters correlate with fibrinogen concentrations [16,38]. Moreover, our study showed the good diagnostic performance of FIBTEM-A10/MCF (AUC = 0.776) in association with hypofibrinogenemia, which corroborates findings from Khunakanan et al. [38]. The initial prevalence of hypofibrinogenemia in trauma patients, as determined by recent studies, was reported to be high, ranging from 7% to 29%, depending on the cutoff value of the fibrinogen concentrations used for diagnosis [38,39,40]. In our study, 24.4% of patients had hypofibrinogenemia upon initial evaluation. Lubkin et al. [39] reported that younger patients with hypofibrinogenemia had greater head AISs and experienced more severe shock and coagulopathy. Additionally, Meizoso et al. [41] reported an association between hypofibrinogenemia and various injury severity scores. Notably, most of our patients were males (91.8%) who were younger, with an average age of 36 years. This distribution reflects the general nature of admitted trauma patients in Qatar [42]. However, it is important to recognize that low fibrinogen levels can contribute to the prolongation of PT, INR, and ROTEM EXTEM-CT. Therefore, the isolated prolongation of these parameters should not be interpreted independently or automatically prompt the administration of FFP or PCC, especially in the absence of active bleeding. Clinical decision-making should be guided by a combination of ROTEM parameters, including FIBTEM A10 (or A5) and plasma fibrinogen concentration, to more accurately identify the underlying coagulopathy and tailor hemostatic therapy accordingly.

We currently follow a ROTEM-guided management protocol for trauma at our institution (Figure 3); however, physician compliance needs to be evaluated in future research and audits. Different institutions across the world have different triggers for their MTP. There is a global trend for each institution to develop and adopt its own MTP in accordance with local resources. Like most MTPs worldwide, our MTP is activated by the clinician in charge of resuscitation in the presence of hemodynamic instability, which does not respond to resuscitation efforts or respond only transiently. While the protocol encourages physicians in charge to consider all clinical (particularly response to resuscitation efforts) and laboratory findings before initiating MTP, the decision ultimately rests on clinical judgement and may be made independently of ancillary tests, including ROTEM. To our knowledge, most MTPs are not initiated solely based on abnormal ROTEM results without hemodynamic instability. However, abnormal ROTEM results are widely used to guide the resuscitation and management of TIC. ROTEM-guided resuscitation has been associated with improved outcomes.

ROTEM is widely used in our Trauma Center for the diagnosis and management of TIC, but it does not trigger the MTP. Our MTP mandates ROTEM once the protocol is activated. Activation of the MTP at our institution is a clinical decision. ROTEM is mandated once the protocol is activated and constitutes a major component of the MT protocol once activated. Notably, ROTEM is also performed even when the MTP is not activated at our institution and is widely used for the diagnosis and management of TIC. However, the use of this protocol in patients who may have massive bleeding and impending shock but early normal parameters remain a challenge. Additionally, patients with minor injuries based solely on the ISS could be misdiagnosed. Finally, according to the European Guidelines on Management of Major Bleeding, point-of-care ROTEM testing improves transfusion practices in trauma patients by enabling faster decision-making, initiation of therapies to correct coagulopathy, improved survival, and cost savings [43].

**Limitations**: The retrospective design of this study may introduce bias and statistical errors. While our study offers insights into coagulation abnormalities post trauma, its ability to account for variations in injury timing and blood sampling is limited, potentially introducing heterogeneity in the progression of these abnormalities among patients. The variability in the timing of ROTEM analysis may have been influenced by ongoing resuscitation efforts, which could affect the accuracy of the results. Moreover, we acknowledge the growing use of FIBTEM A5 in contemporary trauma algorithms. Early clot firmness parameters such as A5 offer the advantage of faster turnaround times, allowing more immediate decision-making in goal-directed hemostatic resuscitation. However, our institutional ROTEM algorithm for TIC follows FIBTEM A10/MCF ≤ 7 to detect coagulation abnormalities and we had adhered to this standard in our analysis. We recognize this as a limitation and recommend that future prospective studies incorporate early ROTEM parameters such as A5 to validate their utility in predicting TIC and guiding transfusion strategies more effectively. Additionally, the correlation between plasma fibrinogen concentration and FIBTEM MCF reported in our study is notably weaker than that reported in other trauma studies. This variability could be partially attributed to the methodological variation in the measurement of ROTEM parameters and plasma fibrinogen concentration. We did not assess pre-existing comorbidities or therapies that could influence acute coagulopathy. Information on the methods used to obtain blood samples during emergency care, especially in patients with severe trauma, resulted in incomplete data on fibrinogen concentrations. Moreover, the INR has a limited effect on the TIC because of its low sensitivity for detecting early or complex coagulopathy. Finally, some patients may have received fibrinogen concentrate, which could affect both the Clauss assay and ROTEM measurements. However, we do not anticipate this significantly affecting our findings, as we believe it to be administered after initial blood investigations. While fibrinogen is crucial for clot formation, abnormal ROTEM findings can be influenced by various factors. Therefore, fibrinogen activity should be interpretated alongside ROTEM analysis for accurate diagnosis and management of coagulopathies. TIC definitions lack consensus, with studies using diverse ROTEM thresholds (e.g., A5, A10, and A15) and PT/INR as an imperfect reference standard. Our study employed a broad definition of ROTEM-assessed TIC. The study by Bonet et al. [19] defined hypocoagulability using ROTEM parameters, considering ATC present in any patient with a ROTEM profile deviating from the normal range. This classification closely aligns with our approach.

## 5. Conclusions

This study revealed that one-fourth of trauma patients had early TIC, as assessed by ROTEM. Patients with altered ROTEM profiles had greater injury severity, greater hypofibrinogenemia, more blood transfusions, and greater mortality. Early ROTEM assessment could lead to goal-directed hemostatic interventions. Additionally, FIBTEM-A10/MCF may serve as a surrogate marker of fibrinogen concentrations. Several studies have previously demonstrated the association between ROTEM parameters and injury severity in trauma patients. Our findings support this relationship and further highlight the clinical relevance of early ROTEM alterations across a broad spectrum of trauma severity. However, given the variability in trauma systems, patient populations, and institutional protocols, further prospective, multicenter studies are warranted to validate and refine the role of ROTEM—particularly in the context of its integration into evolving trauma resuscitation algorithms and decision-making processes.

## Figures and Tables

**Figure 1 diagnostics-15-01148-f001:**
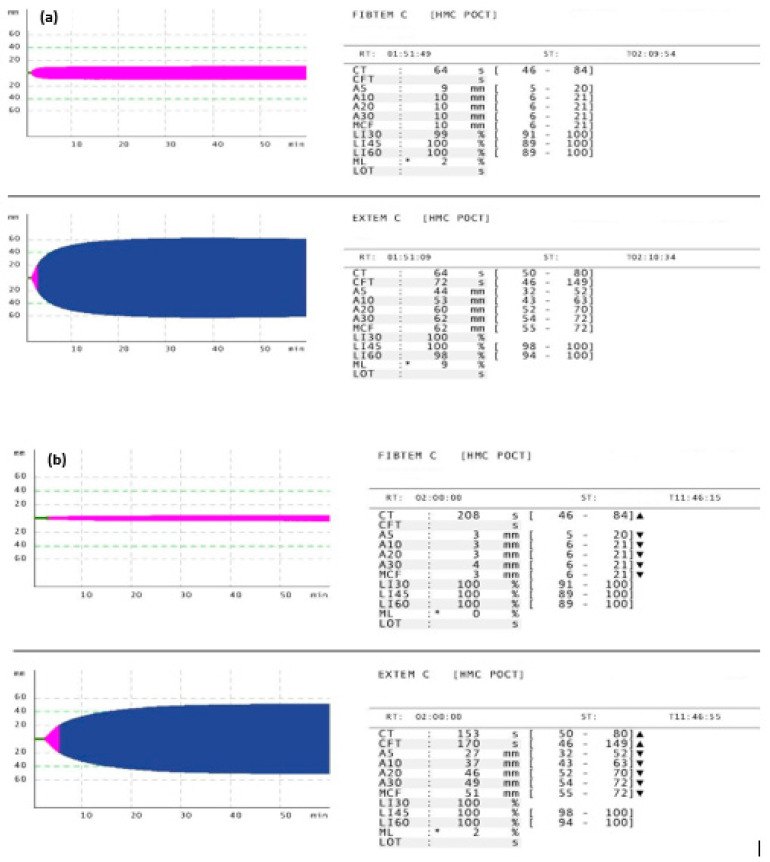
Example of ROTEM (FIBTEM and EXTEM) parameters in trauma patients. (**a**) Normal ROTEM findings. (**b**) Abnormal ROTEM findings. * *Signifies no maximum lysis (ML above 15 indicate lysis).*

**Figure 2 diagnostics-15-01148-f002:**
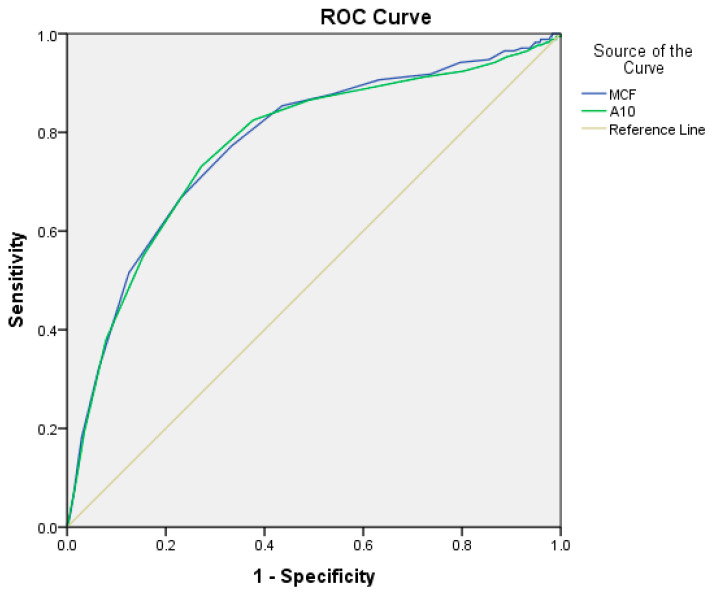
ROC curve for the association of hypofibrinogenemia (defined as plasma fibrinogen concentration < 2.0 g/L) with FIBTEM-MCF and FIBTEM-A10. The area under the curve (AUC) for FIBTEM-MCF was 0.776 (95% CI: 0.735–0.818; *p* = 0.001) with a cutoff value of 9.5 mm (sensitivity: 77.2%; specificity: 66.7%). For FIBTEM-A10, the AUC was 0.772 (95% CI: 0.730–0.815; *p* = 0.001) with a cutoff value of 8.5 mm (sensitivity: 73.1%; specificity: 72.8%).

**Figure 3 diagnostics-15-01148-f003:**
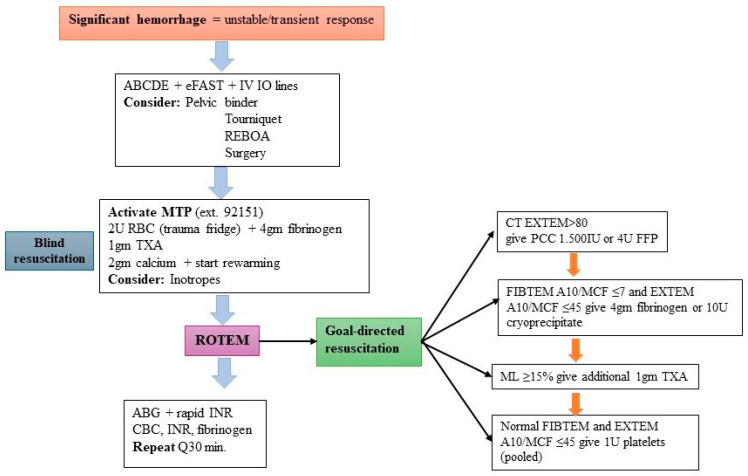
ROTEM algorithm for trauma-induced coagulopathy at the Hamad Trauma Center.

**Table 1 diagnostics-15-01148-t001:** Overall trauma patients with ROTEM findings (*n* = 1488).

Variable	Value	Variable	Value
**Age (mean ± SD)**	36.4 ± 14.2	EXTEM-CT * (50–80 s)	62 (57–66)
**Males**	1366 (91.8%)	EXTEM-CFT * (46–149 s)	72 (61–87)
**Blunt trauma**	1432 (96.2%)	EXTEM-A10 * (43–63 mm)	55 (51–59)
**Vital sign in ED**		Abnormal EXTEM -A10 (*n*; %)	187 (12.5%)
Pulse rate	88.1 ± 18.2	EXTEM-MCF * (55–72 mm)	64 (61–68)
Systolic blood pressure	130.5 ± 22.8	Abnormal EXTEM-MCF (*n*; %)	165 (11.0%)
Diastolic blood pressure	80.4 ± 14.8	EXTEM-ML * (%)	2 (0–6)
Respiratory rate	19.8 ± 4.9	**Overall abnormal EXTEM**	309 (20.8%)
**Shock index**	0.71 ± 0.3	**Normal ROTEM**	1112 (74.7%)
**Glasgow coma score at ED**	15 (3–15)	**Overall abnormal ROTEM**	376 (25.3%)
GCS ≤ 8	114 (7.7%)	**Blood transfusion**	202 (13.6%)
**Injury severity score**	11.5 ± 8.6	**Blood units (median, range)**	4 (1–42)
ISS ≥ 16	352 (23.9%)	**RBC units (<24 h)**	2 (1–21)
**Revised trauma score**	7.5 ± 1.0	**FFP units (<24 h)**	4 (1–26)
**Head AIS**	3.3 ± 0.9	**Platelet units (<24 h)**	6 (2–18)
**Chest AIS**	2.6 ± 0.8	**Massive blood transfusion**	28 (1.9%)
**Abdomen AIS**	2.5 ± 0.9	**Procedures**	
**Pelvis AIS**	2.1 ± 0.5	Open Reduction Internal Fixation	262 (17.6%)
**Prehospital fluids (mL)**	500 (50–2500)	Intubation	201 (13.5%)
**Routine laboratory findings at ED**		Chest tube insertion	90 (6.0%)
WBC * (*n* = 1457) (4–10 × 10^3^/μL)	13.7 (10.1–17.6)	Exploratory laparotomy	52 (3.5%)
Hemoglobin (*n* = 1457) (13–17 gm/dL)	14.1 (12.8–15.1)	Crainotomy/Crainectomy	47 (3.2%)
Hematocrit (*n* = 1455) (40–50%)	42.3 (38.6–44.9)	Thoracotomy	14 (0.9%)
Platelet count (*n* = 1441) (150–400 × 10^3^/μL)	249 (208–294)	**Drug administration**	
Prothrombin time (*n* = 1407) (9.7–11.8 s)	11.4 (11–12.1)	Tranexamic acid	103 (6.9%)
International Normalized Ratio > 1.2	76 (5.4%)	Vasopressors	75 (5.0%)
aPTT (*n* = 1404) (24.6–31.2 s)	24.9 (23.3–26.8)	Fibrinogen	60 (4.0%)
Fibrinogen (*n* = 717) (1.7–4.2 gm/L)	2.5 (2.0–3.0)	Prothrombin complex	5 (0.3%)
Hypofibrinogenemia	175/717 (24.4%)	**In-hospital complications**	
Troponin T (*n* = 1157) (3–15 ng/L)	6 (4–11)	Ventilator-associated pneumonia	29 (1.9%)
Blood pH (*n* = 1296) (7.35–7.45)	7.35 (7.31–7.38)	Sepsis	4 (0.3%)
Lactate (*n* = 1293) (0.5–2.2 mmol/L)	1.9 (1.3–2.7)	Acute respiratory distress syndrome	4 (0.3%)
Base excess (*n* = 1288) (−2 to 2 mmol/L)	−1.2 (−3.3–0.8)	Multiorgan failure	1 (0.1%)
**ROTEM parameters at ED**		**Mechanical ventilation**	200 (13.4%)
FIBTEM-CT, (46–84 s)	65 (61–71)	**Ventilatory days**	3 (1–97)
FIBTEM-A10 * (6–21 mm)	10 (8–13)	**ICU length of stay** (days)	3 (0.04–74)
Abnormal FIBTEM_A10	139 (9.3%)	**Hospital length of stay** (days)	4 (1–366)
FIBTEM ML * (%)	0 (0–0)	**Mortality**	33 (2.2%)
FIBTEM-MCF * (6–21 mm)	10 (8–13)		
Abnormal FIBTEM-MCF	147 (9.9%)		
Overall abnormal FIBTEM	162 (10.9%)		

* Median, IQR; ED: emergency department.

**Table 2 diagnostics-15-01148-t002:** Comparison of clinical characteristics and ROTEM findings based on injury severity score.

	(ISS ≤ 8) *n* = 548	(ISS 9–15) *n* = 588	(ISS ≥ 16) *n* = 352	*p*-Value
**Age (mean ± SD)**	34.4 ± 13.1	38.2 ± 15.5	36.3 ± 13.2	0.001
**Males**	490 (89.4%)	541 (92.0%)	335 (95.2%)	0.009
**Shock Index in ED**	0.66 ± 0.25	0.67 ± 0.24	0.83 ± 0.38	0.001
**Troponin T** (3–15 ng/L)	5 (3–7)	6 (4–10)	8 (5–23.5)	0.001
**Laboratory parameters at ED**				
**Prothrombin time** (median, IQR) (9.7–11.8 s)	11.4 (10.9–11.9)	11.4 (10.9–12.1)	11.9 (11.3–12.7)	0.001
**International Normalized Ratio**	1.0 (1.0–1.1)	1.0 (1.0–1.1)	1.1 (1.0–1.2)	0.001
**aPTT** (24.6–31.2 s)	25.1 (23.3–26.8)	24.8 (23.3–26.6)	24.8 (23.1–27.1)	0.54
**Lactate** (0.5–2.2 mmol/L)	1.7 (1.2–2.4)	1.9 (1.3–2.7)	2.3 (1.5–3.4)	0.001
**Base excess** (−2 to 2 mmol/L)	−0.5 (−2.3–1.4)	−0.9 (−2.8–1.0)	−3.0 (−5.9–1.0)	0.001
**Fibrinogen** (1.7–4.2 gm/L)	2.48 (2.1–2.9)	2.55 (2.0–3.1)	2.35 (1.8–2.9)	0.01
**Hypofibrinogenemia**	44 (18.8%)	61 (21.9%)	70 (34.1%)	0.001
**Hypofibrinogenemia levels ***	1.8 (1.6–1.9)	1.7 (1.5–1.8)	1.6 (1.3–1.8)	0.02
**ROTEM parameters**				
**Abnormal ROTEM**	122 (22.3%)	157 (26.7%)	97 (27.6%)	0.12
**FIBTEM-A10** (6–21 mm)	10 (8–13)	10 (8–13)	9 (7–12)	0.001
**FIBTEM-ML (%)**	0 (0–0)	0 (0–0)	0 (0–0)	0.65
**FIBTEM-MCF** (6–21 mm)	11 (8–13)	11 (8–14)	10 (7–13)	0.001
Abnormal FIBTEM	47 (8.6%)	69 (11.7%)	46 (13.1%)	0.07
**EXTEM-CT *** (50–80 s)	61 (57–66)	62 (57–66)	62 (58–68)	0.004
**EXTEM-A10** * (43–63 mm)	56 (52–59)	55 (52–60)	54 (49–58)	0.001
**EXTEM-CFT** * (46–149 s)	71 (61–85)	71 (59–87)	76 (63–94)	0.002
**EXTEM-MCF** * (55–72 mm)	64 (61–68)	64.5 (61–68)	63.5 (59–67)	0.001
**EXTEM-ML** * (%)	2 (0–5)	2 (0–6)	1 (0–5)	0.12
Abnormal EXTEM	105 (19.2%)	127 (21.6%)	77 (21.9%)	0.50
**Blood transfusion**	12 (2.2%)	72 (12.2%)	118 (33.5%)	0.001
**Massive blood transfusion**	3 (0.5%)	2 (0.3%)	23 (6.5%)	0.001
**Mortality**	1 (0.2%)	1 (0.2%)	31 (8.8%)	0.001

* Median, IQR.

**Table 3 diagnostics-15-01148-t003:** Sub-analysis for moderate and severe traumatic brain injury (TBI).

	Moderate TBI (GCS 9–12) *n* = 39	Severe TBI (GCS ≤ 8) *n* = 114	*p*-Value
**Age (mean ± SD)**	38.1 ± 17.4	34.5 ± 11.6	0.23
**Males**	35 (89.7%)	111 (97.4%)	0.05
**Shock Index**	0.84 ± 0.37	1.04 ± 0.46	0.01
**Troponin T** (3–15 ng/L)	8 (5–27.5)	18 (6–48.5)	0.08
**Fibrinogen** (1.7–4.2 gm/L)	2.4 (1.7–2.5)	2.1 (1.6–2.7)	0.90
**Hypofibrinogenemia**	7 (36.8%)	32 (45.7%)	0.48
**Hypofibrinogenemia levels**	1.5 (1.2–1.8)	1.6 (1.2–1.8)	1.00
**Laboratory parameters at ED**			
**Prothrombin time** (median, IQR) (9.7–11.8 s)	11.9 (11.4–13.4)	12.3 (11.5–13.9)	0.19
**International Normalized Ratio**	1.1 (1.0–1.2)	1.1 (1.1–1.3)	0.09
**aPTT** (24.6–31.2 s)	24.7 (23.5–27.8)	26.3 (23.2–28.5)	0.24
**Serum Lactate** (0.5–2.2 mmol/L)	2.8 (1.7–3.5)	2.9 (1.8–5.1)	0.23
**Base excess** (−2 to 2 mmol/L)	−3.8 (−6.9–1.0)	−6.7 (−8.7–3.0)	0.008
**ROTEM parameters**			
**Abnormal ROTEM**	16 (41.0%)	50 (43.9%)	0.75
**FIBTEM-A10/MCF** (6–21 mm)	9 (6–13)	8 (6–11)	0.24
Abnormal FIBTEM	12 (30.8%)	25 (21.9%)	0.26
**FIBTEM-ML (%)**	0 (0–0)	0 (0–0)	0.20
**EXTEM-CT *** (50–80 s)	63 (59–68)	66 (61–83)	0.03
**EXTEM-A10** * (43–63 mm)	53 (50–59)	51 (46–56)	0.02
**EXTEM-CFT** * (46–149 s)	74.5 (56.3–90.7)	83 (67–114)	0.01
**EXTEM-MCF** * (55–72 mm)	63 (61–67)	62 (57.3–66)	0.05
**Abnormal EXTEM**	11 (28.2%)	40 (35.1%)	0.43
**Blood transfusion**	15 (38.5%)	55 (48.2%)	0.29
**Massive blood transfusion**	3 (7.7%)	13 (11.4%)	0.51
**Mortality**	2 (5.1%)	26 (22.8%)	0.02

* Median, IQR.

**Table 4 diagnostics-15-01148-t004:** Comparison of clinical characteristics and outcomes of patients with and without thromboelastometry (ROTEM) alterations.

	Normal (*n* = 1112)	TIC * (*n* = 376)	*p*-Value
**Age (mean ± SD)**	35.5 ± 12.9	38.9 ± 17.5	0.001
**Males**	1062 (95.5%)	304 (80.9%)	0.001 for all
**Females**	50 (4.5%)	72 (19.1%)	
**Prehospital fluids (mL)**	478.3 ± 357.3	620.1 ± 461.5	0.01
**Laboratory parameters at ED**			
**WBC** (4–10 × 10^3^/μL)	14.3 ± 5.9	14.2 ± 6.2	0.79
**Platelet count** (150–400 × 10^3^/μL)	251.3 ± 63.7	265.8 ± 90.9	0.001
**Prothrombin time** (9.7–11.8 s)	11.6 ± 0.9	12.0 ± 2.1	0.001
**aPTT** (24.6–31.2 s)	25.0 ± 3.6	27.1 ± 11.2	0.001
**International Normalized Ratio**	1.06 ± 0.09	1.12 ± 0.26	0.001
**Lactate** (0.5–2.2 mmol/L)	2.2 ± 1.8	2.6 ± 2.2	0.01
**Fibrinogen** (1.7–4.2 gm/L) *n* = 717	2.6 ± 1.7	2.9 ± 2.3	0.06
Hypofibrinogenemia < 2 g/L	110 (21.0%)	65 (33.7%)	0.001
**Head injury**	245 (22.0%)	98 (26.1%)	0.10
**Head AIS**	3.2 ± 0.9	3.5 ± 1.0	0.04
**Injury severity score**	11.0 ± 8.2	12.6 ± 9.8	0.007
**Shock index**	0.69 ± 0.28	0.75 ± 0.32	0.003
**ROTEM parameters**			
**FIBTEM-A10 **** (6–21 mm)	10 (8–12)	12 (6–17)	0.001
**FIBTEM-MCF **** (6–21 mm)	10 (8–13)	13 (6–19)	0.001
**FIBTEM-ML ** (%)**	0 (0–0)	0 (0–0)	0.29
**EXTEM-CT ****	61 (58–65)	63 (56–71)	0.001
**EXTEM-CFT** ** (46–149 s)	72 (64–85)	64.5 (45–102)	0.001
**EXTEM-A10** ** (43–63 mm)	55 (52–58)	57 (47–65)	0.002
**EXTEM-MCF** ** (55–72 mm)	64 (62–67)	65 (57–72)	0.14
**Blood transfusion**	131 (11.8%)	71 (18.9%)	0.001
**Blood units**	3 (1–42)	4 (1–32)	0.69
**Massive blood transfusion**	15 (1.3%)	13 (3.5%)	0.009
RBC units (<24 h)	85 (7.6%)	58 (15.4%)	0.001
FFP units (<24 h)	20 (1.8%)	17 (4.5%)	0.003
Platelet units (<24 h)	16 (1.4%)	21 (5.6%)	0.001
**Drug administration**			
Prothrombin complex	2 (0.2%)	3 (0.8%)	0.07
Fibrinogen	32 (2.9%)	28 (7.4%)	0.001
Tranexamic acid	69 (6.2%)	34 (9.0%)	0.06
Vasopressors	45 (4.0%)	30 (8.0%)	0.003
**Mortality**	11 (1.0%)	22 (5.9%)	0.001

* Abnormal ROTEM: any single or multiple ROTEM parameters outside the normal range; ** median, IQR; Categorical variables compared using chi-square test.

**Table 5 diagnostics-15-01148-t005:** Comparison of clinical characteristics and outcomes of trauma patient-based initial fibrinogen concentrations *.

	Normal Fibrinogen (>2.0) *n* = 542	Hypofibrinogenemia (≤2.0) *n* = 175	*p*-Value
**Age (years)**	35.9 ± 13.6	32.7 ± 13.1	0.01
**Males**	500 (92.3%)	168 (96.0%)	0.08
**Blunt trauma**	523 (96.5%)	167 (95.4%)	0.52
**Head injury**	140 (25.8%)	52 (29.7%)	0.31
**Injury severity score**	11.7 ± 8.6	15.1 ± 10.8	0.001
**Severe trauma (ISS ≥ 16)**	135 (24.9%)	70 (40.0%)	0.001
**Shock index**	0.72 ± 0.3	0.85 ± 0.4	0.001
**Prothrombin time (median, IQR) at ED**	11.4 (10.9–12.0)	12.1 (11.5–13.0)	0.001
**International Normalized Ratio at ED**	1.0 (1.0–1.0)	1.1 (1.1–1.2)	0.001
**APTT**	24.9 (23.3–26.9)	24.7 (23.1–26.6)	0.56
**Serum Lactate**	2.0 (1.4–2.8)	2.3 (1.5–3.4)	0.17
**Base excess**	−1.2 (−3.4–0.7)	−2.8 (−6.5–0.7)	0.001
**Abnormal FIBTEM**	44 (8.1%)	40 (22.9%)	0.001
**Abnormal EXTEM**	112 (20.7%)	45 (25.7%)	0.16
**Abnormal ROTEM overall**	128 (23.6%)	65 (37.1%)	0.001
**Intubation**	75 (13.8%)	49 (28.0%)	0.001
**Blood transfusion**	71 (13.1%)	54 (30.9%)	0.001
**Massive blood transfusion**	5 (0.9%)	14 (8.0%)	0.001
**Mechanical ventilation**	74 (13.4%)	49 (28.0%)	0.001
**ICU LOS days**	3 (0.04–65)	3 (0.8–74)	0.07
**Hospital length of stay**	4 (1–189)	5 (1–340)	0.002
**Mortality**	10 (1.8%)	10 (5.7%)	0.007

* Admission fibrinogen levels were available in 717 cases.

**Table 6 diagnostics-15-01148-t006:** Discriminatory analysis of ROTEM parameters to predict outcomes.

Parameter	ROC-Derived Cut-Off	AUC (95% CI), *p*-Value	Sensitivity	Specificity	Positive Predictive Value	Negative Predictive Value
**ROC curve for FIBTEM to predict Clauss fibrinogen plasma concentration ≤ 2 g/L**						
**FIBTEM-A10**	8.5	AUC: 0.773; 95% CI: 0.730–0.816, *p* = 0.001	73.1%	73%	10.4% (8.3–13.0%)	53.6% (49.5–57.7%)
**FIBTEM-MCF**	9.5	AUC: 0.777; 95% CI: 0.736–0.819, *p* = 0.001	77.2%	66.9%	9.7% (7.5–12.5%)	57.6% (54.1–61.1%)
**ROC curve for FIBTEM to predict blood transfusion**						
**FIBTEM-A10**	9.5	AUC: 0.572; 95% CI: 0.524–0.620, *p* = 0.001	55.8%	42.7%	10.6% (9.1–12.2%)	83.1% (81.1–84.9%)
**FIBTEM-MCF**	9.5	AUC: 0.559; 95% CI: 0.511–0.608, *p* = 0.007	48.2%	37.7%	11.3% (9.9–12.8%)	83.5% (81.2–85.6%)
**ROC curve for FIBTEM to predict massive blood transfusion**						
**FIBTEM-A10**	9.5	AUC: 0.687; 95% CI: 0.567–0.807, *p* = 0.001	76.9%	56.1%	0.73% (0.3–1.4%)	96.9% (96.2–97.5%)
**FIBTEM-MCF**	9.5	AUC: 0.662; 95% CI: 0.542–0.783, *p* = 0.005	61.5%	61.3%	1.1% (0.6–1.8%)	97.2% (96.2–97.9%)
**ROC curve for FIBTEM to predict in-hospital mortality**						
**FIBTEM-A10**	9.5	AUC: 0.665; 95% CI: 0.533–797, *p* = 0.004	65.4%	55.9%	1.1% (0.6–1.8%)	97.2% (96.4–97.9%)
**FIBTEM-MCF**	9.5	AUC: 0.638; 95% CI: 0.501–0.775, *p* = 0.016	57.7%	61.3%	1.2% (0.8–1.9%)	97.4% (96.4–98.1%)
**ROC curve for EXTEM to predict blood transfusion**						
**EXTEM-CT**	60.5	AUC: 0.525; 95% CI: 0.479–0.571, *p* = 0.257	56.3%	44.3%	13.6% (12.1–15.2%)	86.6% (84.5–88.4%)
**EXTEM-A10**	51.5	AUC: 0.448; 95% CI: 0.402–0.494, *p* = 0.019	65.8%	24.2%	11.8% (10.8–13.0%)	82.0% (78.6–84.9%)
**ROC curve for EXTEM to predict massive blood transfusion**						
**EXTEM-CT**	60.5	AUC: 0.595; 95% CI: 0.468–0.721, *p* = 0.097	65.4%	44.4%	2.1% (1.6–2.8%)	98.4% (97.5–99.0%)
**EXTEM-A10**	51.5	AUC: 0.262; 95% CI: 0.155–0.370, *p* = 0.001	34.6%	24.8%	0.8% (0.4–1.3%)	95.5% (94.0–96.6%)
**ROC curve for EXTEM to predict in-hospital mortality**						
**EXTEM-CT**	60.5	AUC: 0.764; 95% CI: 0.657–0.871, *p* = 0.001	77.4%	50.2%	3.0% (2.5–3.6%)	98.9% (97.9–99.4%)
**EXTEM-A10**	51.5	AUC: 0.239; 95% CI: 0.133–0.346, *p* = 0.001	32.3%	24.6%	0.9% (0.5–1.5%)	94.4% (92.9–95.6%)
**ROC for Clauss Fibrinogen to predict the need for massive blood transfusion**						
Clauss fibrinogen plasma concentration	2.0	AUC: 0.856; 95% CI: 0.785–0.926, *p* = 0.001	73.7%	77.5%	8.0% (6.1–10.5%)	99.1% (98.1–99.5%)
**ROC for Clauss Fibrinogen to predict in-hospital mortality**						
Clauss fibrinogen plasma concentration	2.0	AUC: 0.664; 95% CI: 0.503–0.825, *p* = 0.013	50.0%	76.3%	5.7% (3.69–8.7%)	98.1% (97.1–98.8%)

## Data Availability

All the data are presented in the manuscript, figures, and tables. The device will be available upon reasonable request and after approval from the medical research center of Hamad Medical Corporation after signing a data-sharing agreement.

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
