# Peer review of "Clinical Significance of Rotational Thromboelastometry (ROTEM) for Detection of Early Coagulopathy in Trauma Patients: A Retrospective Study"

_diagnostics, 2025, doi:10.3390/diagnostics15091148_

Round 1

Reviewer 1 Report

Comments and Suggestions for Authors

General Comment: This is an interesting analysis of standard coagulation tests and ROTEM results in a big trauma population (1488 patients). However, the value of the manuscript can be improved significantly by addressing some issues and some additional analyses.

Specific Comments:

Line 2-4 and 27-30 (Abstract): The title “Clinical Significance of Rotational Thromboelastometry (ROTEM) for Detection of Early Coagulopathy in Trauma Patients: A Retrospective Study” suggests that the authors did a comprehensive evaluation of ROTEM assays and parameters to detect early coagulopathy in trauma patients. However, the assessment was limited to FIBTEM MCF, only. This gives rise to the following questions:

Why did you assess FIBTEM, only? TIC has been characterized by EXTEM A5 ≤ 35 mm by Davenport et al. in 2011, and their results have been confirmed by Kelly et al. in 1146 trauma patients (ROC AUC to predict in-hospital mortality for EXTEM A5 , A10 , and MCF was 0.87, 0.89, and 0.90, respectively).

Why did you use the “historic” Parameter MCF for FIBTEM? Modern algorithms use the early clot firmness parameter A5 for the detection of TIC and for guiding therapy. Therefore, reporting FIBTEM A5 results would be very helpful for the reader.

Line 35ff (Abstract): What was the optimum FIBTEM A5/A10/MCF cut-off value to predict a Clauss fibrinogen plasma concentration < 1.5 g/L (2 g/L) and to predict massive transfusion (≥5 units of PRBCs/24 h or ≥10 units of PRBC/24 h) or in-hospital mortality. Please provide ROC AUC, 95% CI, SE, sensitivity, specificity, positive predictive value, and negative predictive value for these endpoints. Please provide the same for Clauss Fibrinogen.

Line 45ff (Abstract): The correlation between plasma fibrinogen concentration and FIBTEM MCF reported in your study is very low compared with other (trauma) studies (Schöchl et al. 2011, r = 0.75; David et al. 2016, r = 0.772-0.858; Mace et al. 2016, R = 0.78; Peng et al. 2018, r = 0.77; Seebold et al. 2019, r = 0.713-0.805; Ziegler et al. 2019, r = 0.8515; Gillissen et al. 2019, r = 0.70 for ROTEM delta and r = 0.85 for ROTEM sigma; de Vries et al. 2020; r >0.8; Ranucci et al. 2020, r = 0.44-0.94; Bell et al. 2025, r = 0.88). How do you explain this very weak correlation observed in your study compared with other studies? Which devices and assays have been used for FIBTEM and Clauss fibrinogen?

Line 49ff (Abstract): Here, the authors state that “additional prospective studies are necessary to further substantiate the association between ROTEM and injury severity in trauma patients”. This relationship has already been assessed in multiple studies. A selection has been listed above.

Line 109ff (Methods): As you refer to the ROTEM sigma reference ranges published by Schenk et al., I assume you used ROTEM sigma for this study. As already discussed above, please provide EXTEM and FIBTEM A5 results, too.

Line 115 (Methods): Why did you look at ML in EXTEM, only. FIBTEM ML is much more sensitive to fibrinolysis compared with EXTEM.

Line 119 (Methods): The manufacturer of ROTEM sigma is TEM Innovations GmbH, Munich. Germany. The device is distributed by Werfen Barcelona, Spain.

Line 125f (Methods): Which device, reagent, and method has been used for measuring the plasma fibrinogen concentration in the lab? The unusual weak correlation between plasma fibrinogen concentration and FIBTEM A10 (MCF) in this study may be due to a high variability of fibrinogen results in the lab

Line 164ff (Results): Please report EXTEM A5, FIBTEM A5 and FIBTEM ML in this table. Notably, EXTEM and FIBTEM cut-off values predictive for bleeding, massive transfusion, and mortality are not identical with the limits of reference ranges. Therefore, I strongly recommend calculating these cut-off values by ROC curve analysis and to replace “abnormal” values with cut-off values. Cut-off values to predict an outcome in a specific clinical setting (such as trauma) provide much more clinical value compared with reference ranges (assessed in health individuals).

Line 213ff (Results): I assume, in this ROC curve analysis hypofibrinogenemia is defined as a plasma fibrinogen concentration < 2 g/L. Since every figure legend should act as a stand-alone, this should be mentioned, here.

Line 45ff (Abstract) and line 150ff (Results): In the abstract, you report that the correlation coefficient between plasma fibrinogen concentration and FIBTEM MCF was only 0.294 (which is very low compared with other studies; see comment above), but I cannot find any correlation results in the results section. Therefore, I am not sure whether the r-value reported in the abstract section (line 45ff) is a typo. Please clarify.

Furthermore, I strongly recommend running ROC curve analyses to predict any transfusion, massive transfusion, and in-hospital mortality based on EXTEM CT, EXTEM A5, EXTEM ML, FIBTEM A5, FIBTEM ML, PT, plasma fibrinogen, platelet count, base excess, and lactate. This would improve the value of the manuscript, significantly, and would provide cut-off values for a local trauma algorithm.

Line 261ff (Discussion): It is important to know that low fibrinogen values can also prolong clotting times (PT/INR as well as ROTEM EXTEM CT). Therefore, prolonged PT (increased INR) and EXTEM CT should only be considered as an indication for FFP or PCC administration in case of bleeding and normal FIBTEM A5 (A10) or normal plasma fibrinogen concentration.

Line 305ff (Discussion): Again, the correlation between the plasma fibrinogen concentration and FIBTEM MCF (r = 0.294 as reported in line 45ff in the abstract) is much weaker compared to any other study assessing this correlation (r = 0.70-0.88; see comment on line 45ff and the corresponding references). If the r-value reported in line 45ff is not a typo, this strong discrepancy must be discussed, here. The weak correlation reported in the abstract, is also not in-line with the good ROC AUC of 0.776 to detect hypofibri.

Author Response

Reply to reviewer's comment attached

Thank you

Reviewer 2 Report

Comments and Suggestions for Authors

Asim M. et al. aim to assess the role of ROTEM in identifying trauma-induced coagulopathy. The timely coagulation defects evaluation in trauma patients is clinically relevant and has a great need in emergency medicine. Although the authors performed a retrospective analysis in a large study cohort, nonetheless, there are several concerns that require additional attention:

  1. It is not clear from the introduction what is the novelty of this manuscript compared to numerous previously published papers.
  2. In the introduction, the authors write “management of hemorrhages and TICs”. However, TICs also include hemorrhage, which the authors do not specify or describe. This comment is also relevant to the Results section, where the authors talk about the abnormal ROTEM parameters. Abnormal means hypo- or hypercoagulation?  
  3. Could you please justify why the pediatric cut-off was <14 years? The pediatric population is considered <18 years.
  4. In the methods, please add references to all the scores that have been used.
  5. To compare categorical variables, the authors should use the Chi-Squared or Fisher’s exact test
  6. Table 1, Abnormal EXTEM 187 – is it a sum of one of the EXTEM parameters or a number of subjects? The column is labeled “value”, but it is unclear what type of value if being referred to. Please, revise the table to enhance its clarity and comprehensibility.
  7. Table 2: 1) Please clarify what was compared to what. There is only one p-value and 3 groups. 2) There are statistically different numbers of males in the group, but is it clinically important? 3) PT, INR fibrinogen, and EXTEM values do not look that much different between the groups and within the normal values. What is the clinical relevance of this statistical difference? 4) Hypofibrinogenemia is reposted in %, but what is more important is the median and the range to characterize its clinical aspect.
  8. In Table 3, hypofibrinogenemia should be compared using actual values rather than percentages, utilizing either the Student's t-test or the Mann-Whitney U test. Additionally, based on the fibrinogen values presented, it does not appear that there were any patients with clinically significant hypofibrinogenemia.
  9. In Table 4, the ages in both groups look comparable; males and females should be evaluated using Chi-Squared.
  10. In the Discussion, the authors address the correlation of ROTEM with other parameters. However, there is no mention of correlation tests in the Statistical approach and no correlation data in the Results. Please, conduct a proper analysis or rephrase the sentences.
  11. Figure 3: Which guidelines were used to assemble the algorithm?

Author Response

Reply to reviewer comments attached

Thank you
